# A Liquid Metal-Enhanced Wearable Thermoelectric Generator

**DOI:** 10.3390/bioengineering9060254

**Published:** 2022-06-14

**Authors:** Wei Liu, Zhenming Li, Yanfang Yang, Chengbo Hu, Zhen Wang, Yongling Lu

**Affiliations:** 1Energy Storage and Novel Technology of Electrical Engineering Department, China Electric Power Research Institute Limited Company, Beijing 100192, China; liuwei3@epri.sgcc.com.cn (W.L.); yangyanfang1@epri.sgcc.com.cn (Y.Y.); 2State Grid Jiangsu Electric Power Co., Ltd. Research Institute, Nanjing 211103, China; 15105168989@163.com (C.H.); wangzhenscut@163.com (Z.W.); 15105182955@163.com (Y.L.)

**Keywords:** liquid metal, phase change material, wearable thermoelectric generator, body heat harvesting, wearable sensor

## Abstract

It is a key challenge to continuously power personal wearable health monitoring systems. This paper reports a novel liquid metal-enhanced wearable thermoelectric generator (LM-WTEG that directly converts body heat into electricity for powering the wearable sensor system. The gallium-based liquid metal alloys with room-temperature melting point (24~30 °C) and high latent heat density (about 500 MJ/m^3^) are used to design a new flexible finned heat sink, which not only absorbs the heat through the solid-liquid phase change of the LM and enhances the heat release to the ambient air due to its high thermal conduction. The LM finned is integrated with WTEG to present high biaxial flexibility, which could be tightly in contact with the skin. The LM-WTEG could achieve a super high output power density of 275 μW/cm^2^ for the simulated heat source (37 °C) with the natural convective heat transfer condition. The energy management unit, the multi-parameter sensors (including temperature, humidity, and accelerometer), and Bluetooth module with a total energy consumption of about 65 μW are designed, which are fully powered from LM-WTEG through harvesting body heat.

## 1. Introduction

The full-solid thermoelectric generator based on the Seebeck effect could directly convert thermal energy into electricity. It owns many advantages, such as silent operation, compactness, and no moving parts [1]. Recently, the flexible wearable thermoelectric generator (WTEG) (as a passive method) has attracted extensive attention due to its unique capacity of harvesting body heat for powering wearable electronics continuously [2]. However, WTEG has a relatively low output power (<10 μW/cm^2^ at free of wind and moveless conditions for most reported work) [3], which limits its application for wearable devices. The WTEG performance is not only dependent on thermoelectric material properties and geometry structure, and also strongly determined by thermal conditions (such as skin temperature and the air-side natural convection) [4,5]. Among these factors, it is crucial to achieving a large temperature difference between the cold/hod sides of the thermoelectric legs for harvesting the body heat [6]. Thus, the performance of WTEG could be remarkably improved by enhancing the thermal release of its cold side.

The room-temperature liquid metal (LM, gallium alloys) has been widely concerned due to its low-temperature melting point, high thermal and electrical conductivities, and non-toxicity [7,8]. LM has been widely applied to biomedical technology [9,10], flexible electronics [11,12,13], soft machines [14,15], high heat-flux thermal management [16,17,18,19], and energy harvesting [20]. The gallium-based LM has been also used as stretchable electrodes to WTEG due to its excellent fluidity [20,21,22]. Suarez et al. [23] applied the LM-based elastic composites and stretchable electrodes to prepare flexible WTEG, which enables a low interconnect resistance as well as excellent stretchability. It could achieve an output power density of 1.75 μW/cm^2^ under the ambient air temperature of 24 °C. Sargolzaeiaval et al. [24] have adopted an LM-based elastomer as a heat spreader, achieving a 1.7-times improvement for the output power. It could obtain high performance of 30 μW/cm^2^ at an air velocity of 1.2 m/s when worn on the wrist. Ramesh et al. [25] filled a low thermal-conductive and flexible aerogel/silicone composite into the gap between the P/N-type thermoelectric legs to enhance the performance of the LM-based WTEG, which achieves an output power density of 5.4 μW/cm^2^ when exposed to the ambient temperature of 24 °C. Lv et al. [3] integrated a copper-foam heat sink with the LM-based elastomer to reduce the thermal resistance at the cold side of WTEG, which could achieve a superhigh output power density of 15.8 μW/cm^2^ at free-moving conditions and 97.6 μW/cm^2^ for walking of 0.8 m/s. Jung et al. [26] have adopted a polymeric hydrogel heat sink for thermal absorption and achieved high performance of 9.7 μW/cm^2^. Lee et al. [27] also used a hydrogel-based heat sink to the LM-based flexible WTEG and obtain an output power density of 8.32 μW/cm^2^. Kim et al. [28] applied a superabsorbent polymer to obtain a large temperature difference between the cold/hot sides of the WTEG. It could achieve 13 μW/cm^2^ when worn on the artificial arm. Lee et al. [29] introduced a PCM (phase change material) as thermal absorption and achieve 20 μW/cm^2^. Hong et al. [30] have designed a high thermal conductive soft electrode to obtain an output power density of 6.96 μW/cm^2^ when harvesting body heat. Khan et al. [31] introduce a radiative cooling method to enhance the thermal release of the cold side of WTEG and obtain an output power density of 12.48 μW/cm^2^ when worn on the wrist.

The literature reviewed above has suggested that the cold-side heat transfer of the WTEG is one of the most important factors in determining its output power. This paper reports a novel LM-enhanced WTEG (LM-WTEG). The flexible LM-finned heat sink is integrated to absorb the heat through the solid-liquid phase change of the LM and enhance the heat release to the ambient air due to its high thermal conduction. In Section 2, the LM-WTEG and wearable sensors are designed and fabricated. In Section 3, LM-WTEG performance and its application for body heat harvesting are investigated in detail.

## 2. Materials and Methods

### 2.1. Materials

The elastomer of Ecoflex 00-30 is provided by Smooth-On. The LMs of pure Ga and Zn are supplied by Zhuzhou Yilong Hung Industrial Co. Ltd. (China). For Ga_96.3_Zn_3.7_ alloy (denoted by GaZn) preparation, they are mixed in a vacuuming oven for 4 h at 150 °C. The P/N-type thermoelectric legs adopted Bi_2_Te_3_ (provided by Changshan Dajiang Electric Appliance Factory, Zhejiang, Quzhou, China). The electrical conductivity is *σ* = 9.3 × 10^4^ S/m, thermal conductivity is *k* = 1.28 W/mK, and Seebeck coefficient is *s* = *s_P_* = |*s_N_*| = 2.02 × 10^−4^ V/K, respectively. The merit of the figure (ZT) of WTEG is then estimated by ZT = s^2^*Tσ*/*k*, which is about ZT = 0.96 at *T* = 300 K.

### 2.2. LM-WTEG Working Principle

Figure 1 shows the LM-WTEG working principle and preparation process. The working principle of the LM-WTEG is shown in Figure 1a. The open output voltage of WTEG is determined by
*V_open_* = *N(s_p_* − *s_N_*)∆*T*(1)
where *N* is the pair number of P/N-type thermoelectric legs, and ∆*T* = *T_h_* − *T_c_* is the temperature difference between the hot/cold sides of the thermoelectric leg [32]. The output power of WTEG is given by
*P* = (*N*(*s_P_* − *s_N_*)^2^Δ*T*^2^)/(4*R_TEG_*)(2)
where *R_TEG_* denotes the single electrical resistance of the P/N-type thermoelectric legs. Thus, ∆*T* is the determining factor of the output power for the given WTEG. When the WTEG is tightly attached to the skin, the hot-side temperature *T_h_* approaches the skin temperature. When the cold side of WTEG is directly exposed to the ambient air, however, the cold-side temperature *Tc* deviates from the air temperature due to the low natural convection coefficient [3]. It would lead to a low ∆*T* and reduce the WTEG performance. Here, the flexible LM-finned heat sink is used to enhance the thermal release from the skin through WTEG. The melting temperature (*T_m_*) of the LM is designed as 24 °C~30 °C, which is higher than the ambient temperature and lower than the skin temperature. The integrated LM-finned heat sink would lead to a large ∆*T* = *T_h_* − *T_m_* (*T_c_* = *T_m_*) due to the thermal absorption from the solid-liquid phase change of the LM. In addition, the high heat conductivity of the LM would enhance thermal release to the ambient air.

### 2.3. LM-WTEG Fabrication

The LM-WTEG preparation process mainly includes the fabrication of a flexible LM-finned heat sink and WTEG. Figure 1c illustrates the fabrication process of the LM-finned heat sink through the cast molding method. The water-solubility polyvinyl alcohol (PVA, PolyDissolveTM-S1, provided by Polymaker) template model with a rectangular slot array (with the size of 3 mm × 0.8 mm × 4 mm, space between slots about 1.8 mm) is prepared through the 3D printer (Raise3D N2). The LM is filled into rectangular slots through vacuuming, which is then frozen at a temperature of −60 °C. The polyimide (PI) film (thickness of 30 μm) integrated with copper electrodes (with the size of 0.25 mm × 1.3 mm × 3.9 mm, space between electrodes about 1.8 mm) is used to weld with LM slots. The LM fin array is obtained after dissolving the PVA template model. The LM composite is prepared by mixing Ecoflex00-30 A and B(Smooth-On) with the LM. Another PVA-based template model with a rectangular slot array (with the size of 3 mm × 1.6 mm × 5 mm) is applied to cover the LM fin array. The uncured composite is then filled into the gap between the LM slots and mold, which is then cured at room temperature for 24 h. The LM-finned heat sink is then achieved by coating the LM elastomer composite and removing the template model.

The FPCB-based WTEG preparation is presented in Figure 1d. The copper electrode (with the size of 0.25 mm × 1.3 mm × 3.9 mm, space between electrodes about 1.8 mm) array is integrated onto PI film with a thickness of 30 μm through the FPCB method, which is used as a flexible substrate. The solder paste (Sn96.5Ag3.0Cu0.5) is printed onto the copper electrodes through the mesh mask (with the slot size of 0.1 mm × 1.3 mm × 1.3 mm). We use surface-mounted technology (SMT, CHM-T560P4, by Charmhigh) to arrange the P/N-type thermoelectric legs (with the size of 3.0 mm × 1.3 mm × 1.3 mm, total pair number of 84) on the solder paste layer. The solder pastes and copper electrodes are positioned on the thermoelectric legs in turn through SMT, which is then placed into the reflow oven for soldering. The LM-WTEG is then integrated by assembling the LM-finned heat sink onto the WTEG with the thermal conduction glue (see Figure 1e). The LM-WTEG presents biaxial-bending performance due to the flexible substrate and LM fin array.

### 2.4. Energy Management and Low-Power Sensors Design

The WTEG output voltage is often unstable and lower than 1 V, which must be boosted to stable and large output (2~5 V) for charging the battery or powering the sensors. According to the previous work [3], we adopted a combined DC-DC converter based on BQ25507 (Texas Instruments, Dallas, TX, USA) and LTC3108 (Linear Technology, Milpitas, CA, USA) (see the circuit Figure A1 in Appendix A). It could achieve a high DC-DC conversion efficiency (such as about 50% for an input voltage of 100 mV), and work at a low input voltage of 20 mV. The low-power sensors of humidity/temperature (SHT35, Sensirion, Zurich, Switzerland, 1.64 μW for frequency of 0.1 Hz) and accelerometer (LIS3DH, Adafruit, New York, NY, USA, 19.8 μW for frequency of 25 Hz). The Bluetooth module (nrf52832, Nordic semiconductor, Oslo, Norway, 43.56 μW for frequency of 1 Hz) is used to collect data from sensors. The total energy consumption is about 65 μW. The mobile phone could receive these data for analysis through a Bluetooth module (see Figure 1b).

### 2.5. Characterization Method

The LM solid-liquid phase change behavior is characterized by the differential scanning calorimetry method (DSC, 200F3Maia, NETZSCH, Selb, Germany). Samples with a mass of 20 mg were put into aluminum pans to cool and heat in the temperature range of −40 °C to 60 °C. Nitrogen was passed in as sample purge gas and guard gas with 20 mL/min. At the same time, the impacts of different cooling/heating rates of 10 °C/min and 15 °C/min on LM onset melting point, onset solidifying point, and fusion enthalpy were studied, respectively.

The heat conductivity of the LM composite is measured by the transient plane source method (TPS, 2500S, Hot Disk). The test sample was a cylinder with a height of 10 mm and a diameter of 40 mm measured by the 5501-type microprobe (radius 6.043 mm). Each sample is measured at least 5 times, and the average thermal conductivity value is considered.

The microstructure is obtained by a Hitachi SU3500. For the LM-WTEG performance test, we used T-type thermocouples (the accuracy of 0.3 °C) to record the temperatures (collected through the Agilent 34970A Data acquisition instrument). In addition, the Voltage-Current (V-I) curves are collected by Keithley 2450 source measuring unit.

## 3. Results and Discussion

### 3.1. Thermal Performance of the LM-Finned Heat Sink

Figure 2 shows the thermal properties of the LM and its elastic composites. The Ga (or GaZn) mixtures are composed of 80wt% Ga (or GaZn) in the elastomer of Ecoflex00-30, which is applied to coat the LM slot and form the elastic fin. As displayed in Figure 2a, the elastomer base has a low thermal conductivity of 0.2 W/mK, which would hinder the thermal release to the ambient air. It could greatly improve the thermal conduction of the elastomer to 0.906 W/(m·K) by 353% for GaZn mixture and 0.916 W/(m·K) by 358% for Ga mixture, respectively. It is noteworthy that adding the LM droplets (about an average diameter of 20 μm) into the elastomer cannot weaken its flexibility due to the liquid state [18]. DSC curves of the Ga and GaZn are shown in Figure 2b. The results indicate that the onset melting temperatures of Ga and GaZn are not affected by the ramp rate. However, the onset solidifying point of the LM is strongly dependent on the ramp rate. Figure 2c,d summarize the thermal properties of the LM. GaZn has a melting temperature of 24.62 °C and a high volumetric fusion enthalpy of 538.6 J/cm^3^, and 30.08 °C and 487.4 J/cm^3^ for pure Ga, respectively. Thus, the high phase-change latent heat of the LM-finned heat sink can effectively absorb the heat of the cold side of WTEG and enhance its output power. The effects of melting temperature (24.62 °C for GaZn and 30.08 °C for Ga) are also considered here.

### 3.2. Performance of the LM-WTEG

Figure 3 shows the impacts of the LM-finned heat sink on the performance of the WTEG. As presented in Figure 3a, all the tests are conducted by placing the LM-WTEG on the hot plate, which is assumed as the heat source. The cold side of the LM-WTEG is directly exposed to the ambient air (about 20 °C). There are four types of LM fins, including Ga or GaZn fins (composed of the Ga/GaZn slots and the coated layer of Ga/GaZn elastic composites) and Ga or GaZn mixtures (composed of Ga/GaZn-80wt% elastic composites). The hot plate temperature is set as 37 °C to match the human skin temperature. It is found in Figure 3b that the temperature difference (∆*T*) between the cold/hot sides of the thermoelectric legs falls rapidly for Ga or GaZn mixtures without phase change due to a large undercooling degree of LM droplets. It would lead to a rapid decrease in the LM-WTEG output power. However, the Ga (or GaZn) fins could maintain a large ∆*T* during its phase change, such as Δ*T* = 5.9 °C for Ga and Δ*T* = 9.2 °C for GaZn, which thus enables high output power of 3.3 mW and 1.1 mW for fin height of 2 mm, respectively. When the heat source temperature (such as 37 °C) is higher than the melting temperature of Ga (30.08 °C, or 24.62 °C for GaZn), the thermal energy is passed through the thermoelectric legs and absorbed by the LM-finned heat sink. It would induce the phase change of Ga (or GaZn) and keep the relatively constant temperature on the cold side of the legs. It is noteworthy that a larger output power would be obtained for GaZn fins compared with the Ga-based case due to its lower melting temperature. As presented in Figure 3c, the GaZn-finned heat sink could achieve an average output power density of 275 μW/cm^2^ during phase change, and 92 μW/cm^2^ for the Ga-based case (the total area of the LM-WTEG is about 12 cm^2^).

When the fin height of 2 mm increases to 4 mm, the phase-change duration would be extended to enhance the performance of the LM-WTEG, as shown in Figure 3d–f. For the GaZn fin with a height of 4 mm, the phase change time is 355 s, which increases 135% compared with that of 2 mm (151 s). Similar results are also found for the Ga-based case, such as the output power of 1.1 mW with a duration time of 270 s for the height of 2 mm, and 1 mW and 732 s for the fin height of 4 mm, respectively. It is interesting to find that the output powers are nearly equal for the cases with the fin heights of 2 mm and 4 mm during the phase change due to the relatively constant temperature difference of ∆*T*.

The impacts of heat sources on LM-WTEG are studied by changing the surface temperature of the hot plate from 37 °C to 42 °C. Figure 3g–i show the output power, temperature difference of ∆*T*, and the duration time of the phase change for the Ga/GaZn finned heat sink with the height of 4 mm. When the temperature of the heat source is 42 °C, the output power increases to 5.8 mW (483 μW/cm^2^), which is 1.8 times that of 37 °C. Similarly, the output power increases from 1 mW at 37 °C to 2.4 mW at 42 °C with an increase of 140%. The main reason is that the temperature difference between the cold/hot sides of the LM-WTEG has increased from 7.5 °C to 13.5 °C. It could be concluded that increasing the temperature of the heat source can effectively increase ΔT, and thus largely improve the output power. It is also noteworthy that the duration time of the large output power would decrease when the heat absorption rate of PCM increases because of the heat source temperature increase.

Figure 4 presents the impacts of wind speed on the LM-WTEG performance. A fan is placed on top of the LM-WTEG to stimulate the ambient airflow. For the GaZn-finned (h = 2 mm) WTEG, a wind speed of 0.4 m/s (corresponding to the walking speed) would enable a high output power of 4.08 mW (340 μW/cm^2^) with a duration time of 156 s (see Figure 4a,b). Its output power has an increase of 24% compared with the case of 0 m/s, while the duration time is almost equal. Similar results are also observed for the fin height of 4 mm, the output power is 4.1 mW (342 μW/cm^2^) with a duration time of 357 s. These results indicate that the phase change of the LM-finned heat sink is mainly determined by the heat source. The main reason lies in that the melting temperature (24.62 °C) of GaZn is close to the ambient air temperature of 20 °C, and deviates from the heat source temperature (37 °C). Thus, enhancing the convective heat transfer of the air side could improve LM-WTEG performance, and be hard to prolong phase change time. Figure 4c,d present the test results for the Ga-finned WTEG. The LM-WTEG with h = 2 mm could achieve an output power of 2.1 mW (175 μW/cm^2^) and a duration time of 542 s for a wind speed of 0.4 m/s, which increases 91% and 88% compared with that of the wind speed of 0 m/s, respectively. When the Ga fin height increases to 4 mm, the output power is 1.6 mW (133 μW/cm^2^) with a duration time of 1045 s for the wind speed of 0.4 m/s. Compared with GaZn, the melting temperature of 30.08 °C is closer to the heat source temperature (37 °C). Thus, the convective heat transfer of the cold air (20 °C) would enhance the thermal release of the fin and prolong its phase change time. When the solid-liquid phase change of the LM fin is fully finished, LM-WTEG could still obtain a high output power (such as 1.9 mW for GaZn) due to its high thermal conductivity (about 30 W/mK for GaZn) [7].

Figure 5 shows the LM-WTEG performance under bent conditions, which is used to simulate the wearing state. When the LM-WTEG is in contact with the curved surface (see Figure 5a), the spacing between the LM fin arrays becomes larger, which is beneficial to enhance the natural convection heat transfer and improve the WTEG performance. To verify the effect of the bending on the LM-WTEG output power, the curved surface with a radius of 55 mm (fixed at a temperature of 37 °C) is selected as the heat source. As shown in Figure 5b, the internal electric resistances of LM-WTEG remain almost the same for the flat and bent states, such as 1.39Ω for the flat state, and 1.40 Ω for the bending along with the direction of the X (or Y) axis. Figure 5c–f present the experimental results of the LM-TEG performance at an ambient temperature of 20 °C. For the GaZn fin with a height of 2 mm, the output power of LM-WTEG in the bending state is 4.2 mW, 27% higher than that in the flat state (3.3 mW), and the time increases by 19% (151 s in the plane state and 180 s in bending state), respectively. When the height of GaZn fin increases to 4 mm, the output power of LM-WTEG in the bending state is 3.8 mW, 18% more than that in the plane, and the time increases by 21%. The results show that bending conditions can improve the performance of LM-WTEG. These results are consistent with the previous work [3].

### 3.3. Application of the LM-WTEG for Body Heat Harvesting

Figure 6 presents the application of LM-WTEG-powered wearable sensors. We integrated LM-WTEG with the energy management unit and sensors, as shown in Figure 6a. The LM-WTEG module (with a total area of 31 mm × 38.8 mm) has 84 pairs of P/N-type thermoelectric legs (with the size of 1.3 mm × 1.3 mm × 3 mm). Figure 6b displays that the LM-WTEG-powered wearable sensor could be tightly in contact with the forehead due to its excellent flexibility. The forehead temperature is about 34.3 °C, which enables LM-WTEG to achieve a high output power of 500 μW at rest (the ambient temperature is about 21 °C). It could supply the designed multi-parameter sensors (65 μW), allowing the sensor to monitor the human body in real-time, including human temperature and humidity (see Figure 6c). The micro accelerometer can monitor human motion states (see Figure 6d). Further, we study in detail the LM-WTEG performance when it is worn on different parts of the human body (such as the forehead, arm, and leg). Figure 6e–j display the average output power and open voltage of LM-WTEG. The LM-WTEG could achieve the average output power of 264 μW When fixed on the arm, and 293 μW for the case of the leg.

## 4. Conclusions

In summary, this paper has reported a novel liquid metal-enhanced wearable thermoelectric. A new flexible finned heat sink based on gallium-based liquid metal alloys with high thermal conduct and latent heat density. It could not only absorb the heat through the solid-liquid phase change of the LM and enhance the heat release to the ambient air. The designed LM-WTEG could achieve a super high output power density of 275 μW/cm^2^ for the simulated heat source (37 °C) under natural convective heat transfer conditions. A low-power (about 65 μW) circuit is also designed, which consists of the energy management unit, the multi-parameter sensors, and the Bluetooth module. When LM-WTEG was worn on the forehead temperature, it achieved a high output power of 500 μW at rest (the ambient temperature of 21 °C), which could power the designed multi-parameter sensors for monitoring the human temperature and humidity and motion states in real-time.

## Figures and Tables

**Figure 1 bioengineering-09-00254-f001:**
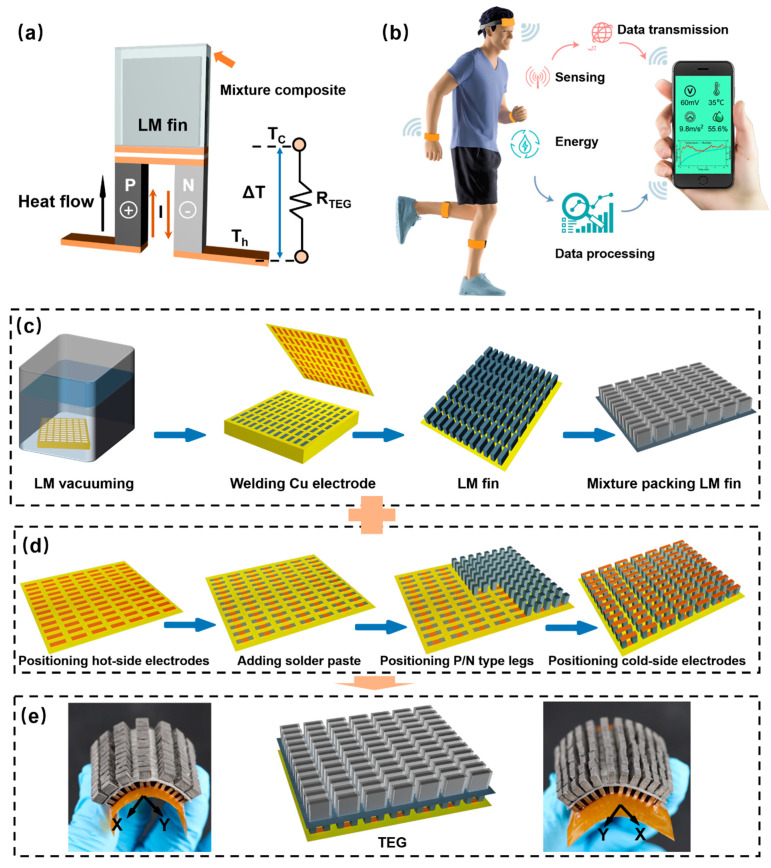
LM-WTEG working principle and preparation process: (**a**) the working principle of LM-WTEG; (**b**) application of LM-WTEG for powering the wearable sensor system; (**c**) flexible LM finned heat sink preparation based on template method; (**d**) WTEG preparation based on flexible printed circuit board; (**e**) biaxial bending of WTEG integrated with LM fin.

**Figure 2 bioengineering-09-00254-f002:**
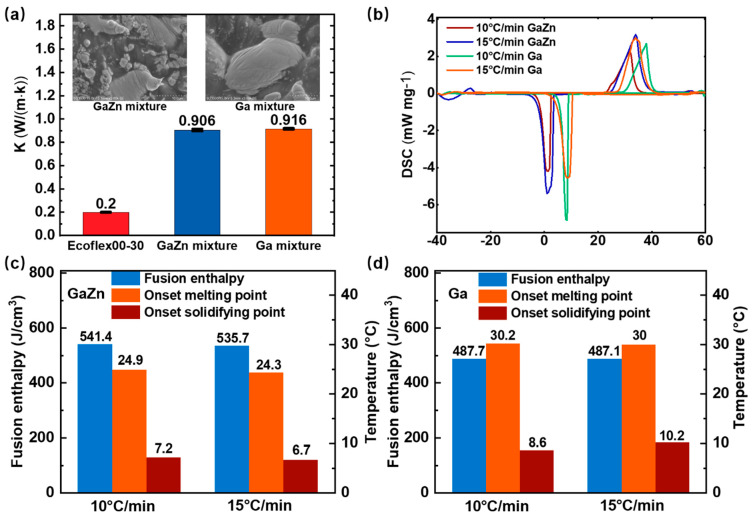
Thermal properties of the LM and its elastic composites LM-WTEG: (**a**) the thermal conductivities of the Ga/GaZn and the mixtures; (**b**) DSC curves of Ga/GaZn; phase change properties of (**c**) GaZn and (**d**) Ga.

**Figure 3 bioengineering-09-00254-f003:**
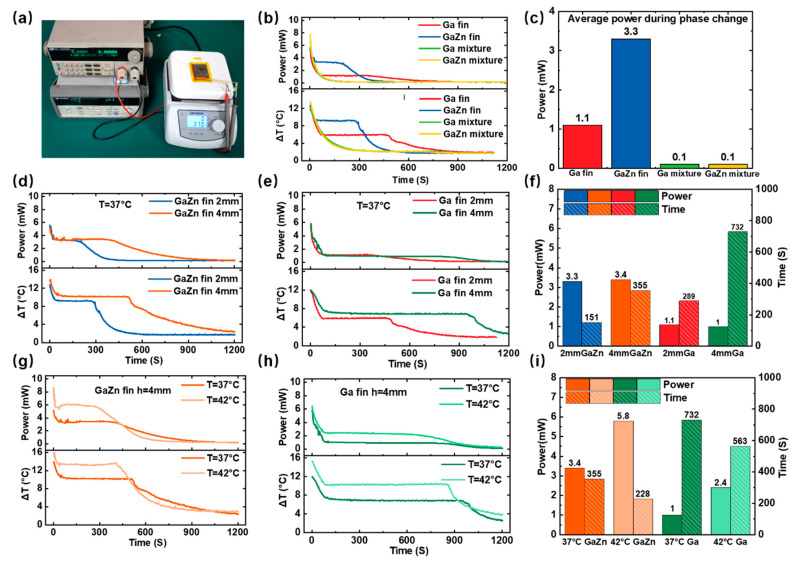
The impacts of LM-finned heat sink on the performance of the WTEG: (**a**) the test platform; (**b**) output power and the temperature ∆*T* of the LM-WTEG for four types of LM fins with the height of 2 mm and (**c**) the corresponding to average outpower during phase change; the impacts of the fin height on the LM-WTEG performance for (**d**) GaZn fin and (**e**) Ga fin, and (**f**) the output power; the impacts of the heat source for the cases of (**g**) GaZn fin and (**h**) Ga fin, and (**i**) the output power.

**Figure 4 bioengineering-09-00254-f004:**
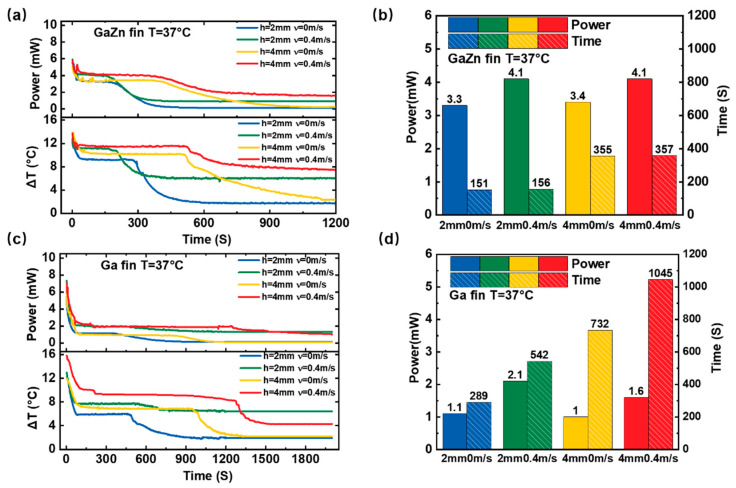
Impacts of the wind speed on the LM-WTEG performance: (**a**) the output power and temperature difference for GaZn fin and (**b**) the corresponding phase change time; (**c**) the output power and temperature difference for Ga fin and (**d**) the corresponding phase change time.

**Figure 5 bioengineering-09-00254-f005:**
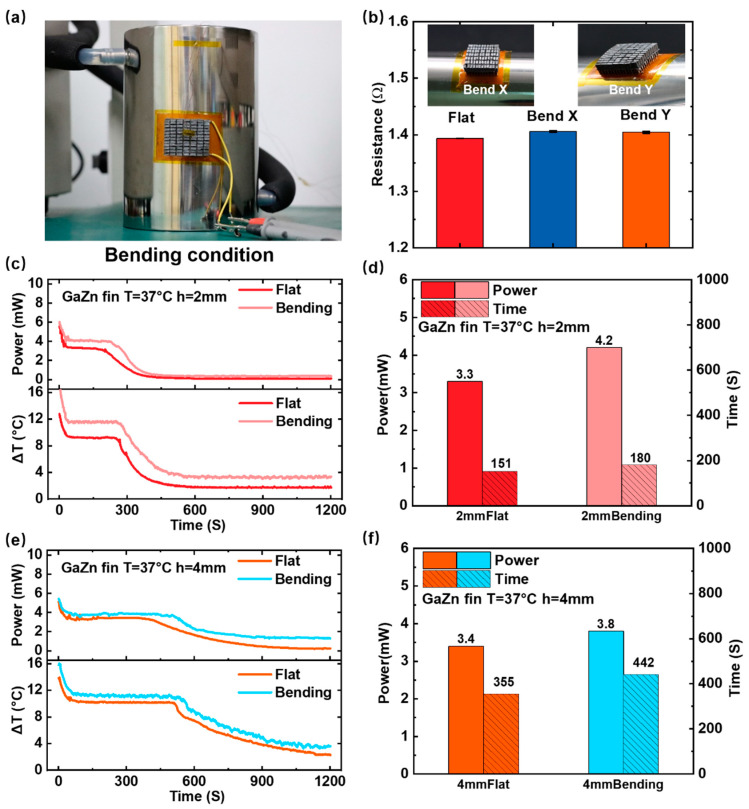
The impacts of the bending on the LM-WTEG performance: (**a**) the test platform; (**b**) the internal electric resistances of LM-WTEG; (**c**) the output power and temperature difference and (**d**) the corresponding phase change time for GaZn fin height of 2 mm; and experiment results of (**e**) and (**f**) for fin height of 4 mm.

**Figure 6 bioengineering-09-00254-f006:**
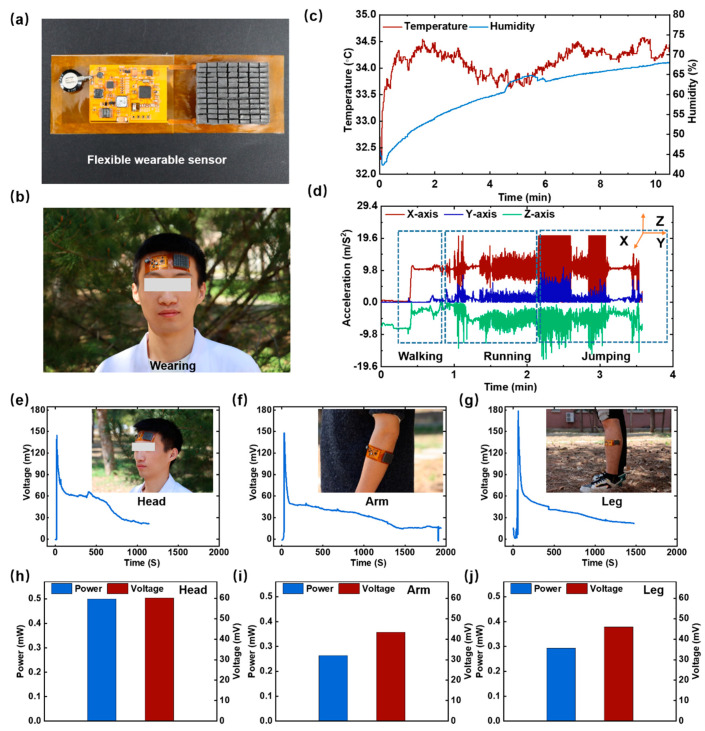
The application of LM-WTEG-powered wearable sensors: (**a**) photos of the wearable sensors powered by LM-WTEG and (**b**) it is worn on the forehead; (**c**) human temperature and humidity data and (**d**) micro accelerometer data; (**e–j**) average output power and open voltage of LM-WTEG when it is worn on the forehead, arm, and leg.

## Data Availability

Data is contained within the article.

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
