# Peer review of "A Liquid Metal-Enhanced Wearable Thermoelectric Generator"

_bioengineering, 2022, doi:10.3390/bioengineering9060254_

Round 1

Reviewer 1 Report

Recently, WTEG for harvesting body heat has been widely concerned, which is expected to be a promising passive method for continuously powering the wearable health monitoring system. The output power of the current WTEG is not maximized yet to meet the supply requirement of the wearable multi-sensor system. This paper reports a novel LM-WTEG, which could achieve a super high output power density of 275 μW/cm2. The main reason lies that the LM not only absorbs the heat through its solid-liquid phase change and enhances the heat release to the ambient air due to its high thermal conduction. This work is very interesting for self-power wearable sensors. The results are also solid. I would like to recommend it published on Bioengineering. The other comments include:

1) Please give the ZT values of the P/N-type thermoelectric legs, which are important for the valuation of the TEG performance

2)     Why is choosing GaZn as LM fins?

3)      Adding the discussion about the impacts of melting temperature on the WTEG performance

Author Response

Recently, WTEG for harvesting body heat has been widely concerned, which is expected to be a promising passive method for continuously powering the wearable health monitoring system. The output power of the current WTEG is not maximized yet to meet the supply requirement of the wearable multi-sensor system. This paper reports a novel LM-WTEG, which could achieve a super high output power density of 275 μW/cm2. The main reason lies that the LM not only absorbs the heat through its solid-liquid phase change and enhances the heat release to the ambient air due to its high thermal conduction. This work is very interesting for self-power wearable sensors. The results are also solid. I would like to recommend it published on Bioengineering.

Re: Thank you for your nice help. We have carefully revised the manuscript.

The other comments include:

1) Please give the ZT values of the P/N-type thermoelectric legs, which are important for the valuation of the TEG performance

Re: Thank you for your nice suggestion. The ZT value of the chosen P/N-type thermoelectric leg is 0.96.

2) Why is choosing GaZn as LM fins?

Re: Thank you for your nice suggestion. GaZn has a melting temperature of 24.62oC and a high volumetric fusion enthalpy of 538.6J/cm3, and 30.08oC and 487.4J/cm3 for pure Ga, respectively. Thus, the high phase-change latent heat of the LM-finned heat sink can effectively absorb the heat of the cold side of WTEG and enhance its output power. The effects of melting temperature (24.62oC for GaZn and 30.08oC for Ga) are also considered here.

3) Adding the discussion about the impacts of melting temperature on the WTEG performance

Re: Thank you for your nice suggestion. It is found in Figure 3(b) that the temperature difference (∆T) between the cold/hot sides of the thermoelectric legs falls rapidly for Ga or GaZn mixtures without phase change due to a large undercooling degree of LM droplets. It would lead to a rapid decrease in the LM-WTEG output power. However, the Ga (or GaZn) fins could maintain a large ∆T during its phase change, such as ΔT=5.9oC for Ga and ΔT=9.2oC for GaZn, which thus enables high output power of 3.3 mW and 1.1 mW for fin height of 2mm, respectively. When the heat source temperature (such as 37oC) is higher than the melting temperature of Ga (30.08oC, or 24.62oC for GaZn), the thermal energy is passed through the thermoelectric legs and absorbed by the LM-finned heat sink.

Reviewer 2 Report

In the manuscript “A Liquid Metal-enhanced Wearable Thermoelectric Generator” by Liu et al, it is demonstrated that the performance of a wearable thermoelectric device can be improved by incorporating highly conductive Ga and GaZn on the heat dissipation side. There is considerable amount of experimental results showing that the proposed method is viable. I recommend the manuscript to be published pending a minor revision addressing the following comments.

My main concern for the manuscript is the lack of discussions. The authors showed that different metals (Ga or GaZn), different configurations (termed as fins or mixtures), sample height (h), etc can influence the outputs of their devices. The underlying reasons, which were not proposed nor discussed, need to be added and discussed to strengthen this work.

The authors are suggested to rationalize why GaZn alloy and its ratio are selected, and explain the differences in fusion enthalpy, thermal conductivity between Ga and GaZn samples.

It is difficult to understand the differences between the sample groups and the terms “fins” and “mixtures” do not seem to describe different samples properly. Please clarify this, preferably together with schematics.

According to the DSC results, the solidification of the samples only occurs when temperature is below about 10 deg C. Therefore, the phase change mechanism will not work in every real-life condition, unless an additional pre-cooling step performed.

Line 175: “The results indicate that 175 the onset melting temperatures of Ga and GaZn are not affected by the ramp rate.” The onset temperature indeed varied considerably in Figure 2b (e.g., for the red and blue curves).

Line 308: a typo here.

Author Response

In the manuscript “A Liquid Metal-enhanced Wearable Thermoelectric Generator” by Liu et al, it is demonstrated that the performance of a wearable thermoelectric device can be improved by incorporating highly conductive Ga and GaZn on the heat dissipation side. There is considerable amount of experimental results showing that the proposed method is viable. I recommend the manuscript to be published pending a minor revision addressing the following comments.

Re: We appreciate very much the reviewer warm-hearted work in help dealing with our paper and giving valuable comments for us to think deeply and further enhance the manuscript. We have carefully studied these comments and tried our best to make suitable improvements. In the following, we wish to explain a little more about the concerns of the reviewer. These points were also incorporated into the revised paper whenever appropriate.

My main concern for the manuscript is the lack of discussions. The authors showed that different metals (Ga or GaZn), different configurations (termed as fins or mixtures), sample height (h), etc can influence the outputs of their devices. The underlying reasons, which were not proposed nor discussed, need to be added and discussed to strengthen this work.

Re: Thank you for your nice suggestion. The cold-side heat transfer of the WTEG is one of the most important factors in determining its output power. The flexible LM-finned heat sink is integrated to absorb the heat through the solid-liquid phase change of the LM and enhance the heat release to the ambient air due to its high thermal conduction. There are four types of LM fins, including Ga or GaZn fins (composed of the Ga/GaZn slots and the coated layer of Ga/GaZn elastic composites) and Ga or GaZn mixtures (composed of Ga/GaZn-80wt% elastic composites ). This paper reports a novel LM-enhanced WTEG (LM-WTEG).

The authors are suggested to rationalize why GaZn alloy and its ratio are selected, and explain the differences in fusion enthalpy, thermal conductivity between Ga and GaZn samples.

Re: Thank you for your nice suggestion. GaZn (Ga96.3Zn3.7) has a melting temperature of 24.62oC and a high volumetric fusion enthalpy of 538.6J/cm3, and 30.08oC and 487.4J/cm3 for pure Ga, respectively. Thus, the high phase-change latent heat of the LM-finned heat sink can effectively absorb the heat of the cold side of WTEG and enhance its output power. The effects of melting temperature (24.62oC for GaZn and 30.08oC for Ga) are also considered here.

It is difficult to understand the differences between the sample groups and the terms “fins” and “mixtures” do not seem to describe different samples properly. Please clarify this, preferably together with schematics.

Re: Thank you for your nice suggestion. There are four types of LM fins, including Ga or GaZn fins (composed of the Ga/GaZn slots and the coated layer of Ga/GaZn elastic composites) and Ga or GaZn mixtures (composed of Ga/GaZn-80wt% elastic composites ). This paper reports a novel LM-enhanced WTEG (LM-WTEG).

According to the DSC results, the solidification of the samples only occurs when temperature is below about 10 deg C. Therefore, the phase change mechanism will not work in every real-life condition, unless an additional pre-cooling step performed.

Re: Thank you for your nice suggestion. There are four types of LM fins, including Ga or GaZn fins (composed of the Ga/GaZn slots and the coated layer of Ga/GaZn elastic composites) and Ga or GaZn mixtures (composed of Ga/GaZn-80wt% elastic composites ). This paper reports a novel LM-enhanced WTEG (LM-WTEG).

Line 175: “The results indicate that 175 the onset melting temperatures of Ga and GaZn are not affected by the ramp rate.” The onset temperature indeed varied considerably in Figure 2b (e.g., for the red and blue curves).

Re: Thank you for your nice suggestion. The true melting point of a substance in a DSC is often considered as the "extrapolated onset temperature" (Please see Ref.: Hohne GWH, et al. The Temperature Calibration of Scanning Calorimeters. Thermochim Acta. 1990(160):1-12.).  All the DSC results presented in manuscript were obtained through automatically calculating by the software from (200F3Maia, NETZSCH). 

Line 308: a typo here.

Re: Thank you for your nice suggestion. We have revised the sentence

Reviewer 3 Report

The paper is on an important topic. It is clearly presented and may be published as is, after only minor correction:

Fig 1(b) - should be "data transmission"  (it is "date")

Fig 1(e) - I cannot see the biaxial bending in those pictures, can you improve?

The formulas in the lines 86 and 88. For better clarity, I suggest putting them as stand out numbered Equation (1) and Equation (2). Resistance T_{TEG} is a resistance of single leg. It is clear from the Fig 1(a), but for a better quality the sentence in line 89 may be like this: 

"""R_{TEG} denotes the electrical resistance of the single P/N-type thermoelectric leg."""

It would also be useful to note below Equation that the power in Eq. (2) is for the matching case when the load resistance is equal to N * R_ {TEG}.

Author Response

The paper is on an important topic. It is clearly presented and may be published as is, after only minor correction:

Re: Thank you for your nice help. We have carefully revised the manuscript.

Fig 1(b) - should be "data transmission"  (it is "date")

Re: Thank you for your nice suggestion. We have revised the word.

Fig 1(e) - I cannot see the biaxial bending in those pictures, can you improve?

Re: Thank you for your nice suggestion. We have added the bending x-y axis in Fig.1(e).

The formulas in the lines 86 and 88. For better clarity, I suggest putting them as stand out numbered Equation (1) and Equation (2). Resistance T_{TEG} is a resistance of single leg. It is clear from the Fig 1(a), but for a better quality the sentence in line 89 may be like this:

"""R_{TEG} denotes the electrical resistance of the single P/N-type thermoelectric leg."""

It would also be useful to note below Equation that the power in Eq. (2) is for the matching case when the load resistance is equal to N * R_ {TEG}.

Re: Thank you for your nice suggestion. We have checked all the equations and descriptions.